# Two-part predictive modeling for COVID-19 cases and deaths in the U.S.

**Teresa-Thuong Le**[1☯], **Xiyue Liao**[2☯]*

**1** California State University, Long Beach, CA, United States of America, **2** San Diego State University, San Diego, CA, United States of America

☯ These authors contributed equally to this work.
* xliao@sdsu.edu

**Data Availability Statement:** All relevant data are located at the following links: https://www.kaggle.com/datasets/johnjdavisiv/us-counties-covid9-weather-sociohealth-data https://www.ncei.noaa.gov/data/global-summary-of-the-day/access/2020/ https://www.atsdr.cdc.gov/placeandhealth/svi/

## Abstract

COVID-19 prediction has been essential in the aid of prevention and control of the disease. The motivation of this case study is to develop predictive models for COVID-19 cases and deaths based on a cross-sectional data set with a total of 28,955 observations and 18 variables, which is compiled from 5 data sources from Kaggle. A two-part modeling framework, in which the first part is a logistic classifier and the second part includes machine learning or statistical smoothing methods, is introduced to model the highly skewed distribution of COVID-19 cases and deaths. We also aim to understand what factors are most relevant to COVID-19's occurrence and fatality. Evaluation criteria such as root mean squared error (RMSE) and mean absolute error (MAE) are used. We find that the two-part XGBoost model perform best with predicting the entire distribution of COVID-19 cases and deaths. The most important factors relevant to either COVID-19 cases or deaths include population and the rate of primary care physicians.

## 1 Introduction

The COVID-19 pandemic continues to upend many people's lives. By 26 March 2020, the United States overtook the world with the most confirmed cases [1]. As of 1 May 2023, there were over 100 million confirmed cases and 1,121,819 deaths reported by the World Health Organization (WHO). Real-time data had been at the forefront during this pandemic. We observed this through COVID-19 forecasting models, which were built using daily-updated data. COVID-19 prediction had not only been trained on real-time data but also, various data sets had been essential in the aid of prevention and control of the disease. Fox et al [2] found that combining both data on COVID-19 deaths and cases and the population's socioeconomic, health, behavioral, and risk factors played an important factor in improving prediction. In another cross-sectional study [3], an adaptive Lasso penalized sliced inverse regression method was used to analyze the relationship between the infection fatality rate of COVID-19 and geographical or demographic features of the infected population in Asia, Europe, Africa, America, and Oceania. This method of variable selection found that whether a patient had cardiovascular disease, the number of physicians, and the number of tests given were significant. These

data_documentation_download.html https://www.
countyhealthrankings.org/explore-health-rankings/
rankings-data-documentation/national-data-
documentation-200-2019 https://www.kff.org/
policy-watch/stay-at-home-orders-to-fight-covid9/
https://github.com/nytimes/covid-9-data.

**Funding:** The author(s) received no specific
funding for this work.

**Competing interests:** The authors have declared
that no competing interests exist.

studies also made use of statistical and machine learning algorithms in predicting the diagnosis of future COVID-19 cases. For example, Fox et al [2] used recurrent neural networks (RNN), more specifically long short-term memory (LSTM) networks, to determine the best predictors for COVID-19 cases. Their integrative deep learning framework could adapt and make predictions based on newly introduced data.

In this case study, we discussed a compiled data set that had 28,955 observations and 18 variables. Outcomes of interest were COVID-19 cases and deaths. 5 data sources from Kaggle were used. The distribution of each outcome was highly skewed with a large positive mass at zero. Such kind of zero-inflated distribution was difficult to fit. Duncan et al's [4] proposed creating a two-part modeling framework combining a logistic regression model with tree-based models such as random forest, gradient-boosted trees, and regularized linear regression. They found that such kind of a framework can fit zero-inflated cost data well. By separating zero-cost patients from patients with positive costs, the model architecture could better capture the skew pattern in the distribution of patient costs than "single-step" models. Distributions of COVID-19 cases and deaths shared a similar skew pattern. In our study, we borrowed this two-part modeling idea to fit the distribution of extremely skewed outcome variables: COVID-19 cases and deaths. In general, a zero-inflated model could be used when the outcome variable had a large mass at zero. However, different models are best for different research goals. The primary research goal in this study was to analyze the non-zero component of COVID-cases or deaths in the data set. The two-part modeling was easy to implement, could separate the zero outcomes and the non-zero outcomes well, and focused on the non-zero outcomes. On the other hand, a zero-inflated distribution could be considered as a mixture of two count distributions, and a zero-inflated model treated zero and non-zero outcomes equally. This two-part modeling framework could be found in recent research about COVID-19 occurrence or fatalities. For example, In Gopal et al's [5], an observational study of all nursing homes in the state of California until 1 May 2020 was done to understand why some nursing homes were more susceptible to larger COVID-19 outbreaks. To model the distribution of COVID-19 cases, a zero-inflated bivariate Poisson regression model was used. Some research about modeling zero-inflated distribution for COVID-19 fatalities was done in Ojinnaka et al's [6]. In this paper, a two-part modeling framework was used to examine the probability of having a COVID-19 fatality and fatalities per 100,000 population in counties with at least 1 fatality in Texas in 2020. A logistic regression model and a multivariate linear regression model formed the two-part model framework. In Li et al's [7], a multivariate two-part model, which is a combination of a logistic regression model and a Poisson regression model, was used to determine the associations of key nursing home characteristics with the likelihood of at least one confirmed case (or death) in the facility, and with the count of cases (deaths) among facilities with at least one confirmed case (death) for all Connecticut nursing homes. However, in these studies, only linear models were considered as part of the two-part modeling framework.

Our goal in this paper was to use a two-part modeling framework to combine logistic regression with statistical non-linear smoothing methods or machine learning models, in order to better handle complex nonlinear relationships between predictors and the outcome of interest. We also aimed to find factors that are most relevant with the outcomes of interest. In this paper, we first explained how we pre-processed the data and explored the data structure in Sections 2.1 to 2.4. In Section 2.5, we presented the information of model building and evaluation. We presented the results of our analysis in Section 3, and closed with a discussion of main findings and limitations in Section 4.

## 2 Materials and methods

### 2.1 Data background

In this study, U.S. Counties: COVID19 + Weather + Socio/Health data set was used, which is available on https://www.kaggle.com/datasets/johnjdavisiv/us-counties-covid19-weather-sociohealth-data. This data set consists of 227 variables with a total of 790,331 observations. It was generated from 5 different data sources. Specifically, the National Oceanic and Atmospheric Administration (NOAA) Global Surface Summary of the Day (GSOD), CDC Social Vulnerability Data, US County Health Rankings, Kaiser Family Foundation, and the New York Times. Out of the 227 variables, there were 18 variables that we were interested in. Labels of the variables were included in Supporting information. From the 2020 NOAA GSOD (https://www.ncei.noaa.gov/data/global-summary-of-the-day/access/2020/) data source, the mean temperature was studied. From the CDC's 2016 Social Vulnerability data (https://www.atsdr.cdc.gov/placeandhealth/svi/data_documentation_download.html), the proportion of Americans living in overcrowding housing and the proportion of Americans living below poverty were included in our study. Other variables of interest were population, rate of primary care physicians, proportion of Americans ages 65 and older, 17 and younger, proportion of each race/ethnicity, and proportion of Americans who live in rural areas. These variables were taken from the 2020 County Health Rankings data source (https://www.countyhealthrankings.org/explore-health-rankings/rankings-data-documentation/national-data-documentation-2010-2019). The Kaiser Family Foundation data source (https://www.kff.org/policy-watch/stay-at-home-orders-to-fight-covid19/) provided state-level stay-at-home orders. Lastly, the count of confirmed COVID-19 cases and deaths due to COVID-19 at a county-level were from the New York Times (https://github.com/nytimes/covid-19-data). We note that there were peculiarities within the data set. The New York Times changed the way they counted deaths in New York City in response to how the state reported its data. The 5 boroughs of New York City (New York, Kings, Queens, Bronx, and Richmond) were combined into one single area. Kansas City had the same peculiarity. Note that the CDC and Health County Ranking data were only available at the county level. Thus, these values were combined for these areas. For New York City and Kansas City, absolute measures, such as population, were summed. This data set did not contain information for every single county in the United States. A county was only included by the New York Times once it reported at least one case.

### 2.2 Skew pattern in the outcome of interest

Table 1 provides summary statistics indicating the extreme skew pattern in the outcome of interest: cases and deaths. $q_\alpha$ is the $\alpha^{th}$ quantile of the variable. The gap between the median value and the maximum value showed an extreme right skew pattern. In addition, we provided histograms for visualization in Fig 1. We set cutoff points to be 100,000 and 5,000 when creating the cases and deaths histograms, respectively. We chose not to include values that were more extreme than the cutoff points in order to better visualize the distributions. There were 43 observations above 100,000 for cases. For deaths, there were 17 observations greater than 5,000.

**Table 1. Cumulative COVID-19 cases and deaths summary statistics.**

| Variables | Min | Mean | $q_{0.25}$ | $q_{0.50}$ | $q_{0.75}$ | $q_{0.95}$ | Max |
|---|---|---|---|---|---|---|---|
| Cases | 1 | 1,890 | 29 | 220 | 999 | 3,347 | 401,034 |
| Deaths | 0 | 52.37 | 0 | 4 | 21 | 70 | 24,323 |

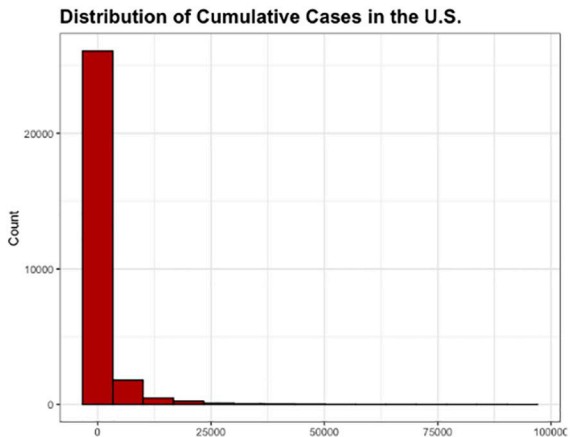
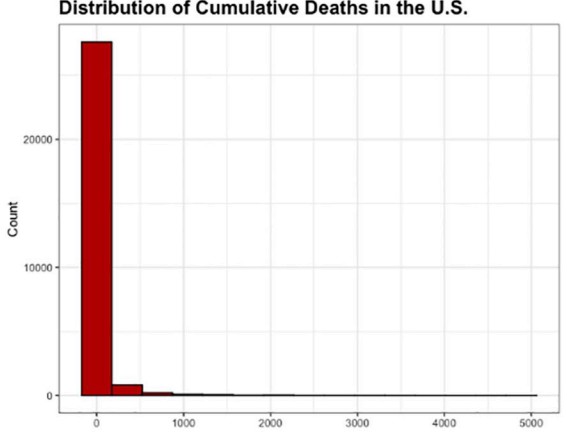

**Fig 1. Distribution of response variables.** Left: distribution of cases. Right: distribution of deaths.

## 2.3 Correlation among continuous variables

The left panel of Fig 2 is the correlation matrix that shows the pairwise Pearson correlation coefficient among 16 continuous variables. Correlations with a $p$-value larger than the significance level 0.05 were marked with a cross sign. The **R** package **ggcorrplot** was used here [8]. We observed that the correlations among the total number of deaths ($x_{16}$), the total number of positive COVID-19 cases ($x_{15}$), and population ($x_1$) were positive and strong, i.e., larger than 0.5. Percentage of non-Hispanic White residents ($x_{10}$) were negatively correlated with percentages of African Americans ($x_5$) and Hispanic ($x_9$) residents. Besides, the percentage of non-Hispanic White residents was less likely to live in an overcrowding area. The right panel is the partial correlation matrix generated by the **ggm** [9] package and the **ggcorrplot** package. Unlike Pearson correlation, partial correlation measures pairwise linear relationship after controlling for other variables. We observed several very strong partial correlations, which

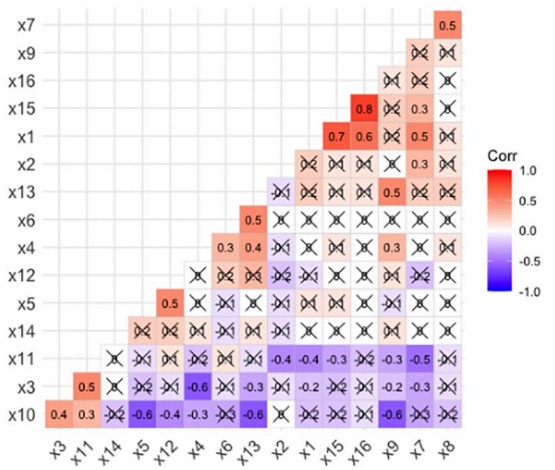

(a) Correlation Matrix

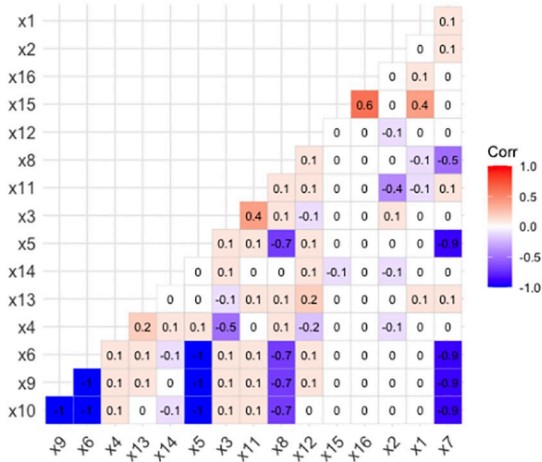

(b) Partial Correlation Matrix

**Fig 2.** Left: Pearson correlation matrix of continuous variables. Right: partial correlation matrix of continuous variables.

especially appeared among racial percentages. For example, the partial correlations between percentages of African Americans ($x_5$) and percentages of Hispanic ($x_9$) residents, percentage of Indian-Americans and Native Alaskans residents, and percentage of non-Hispanic White residents ($x_{10}$). The labeling of all variables in this matrix could be found in S1 and S2 Tables.

We used the vif function in the **car** package [10] to compute the generalized variance-inflation factor (GVIF) [11] among all predictors. In the **car** package, generalized variance-inflation factors were computed when some term had more than 1 degree of freedom, which was the case of this study because the predictor "state" had 50 degree of freedoms. After adjusting for the degree of freedom, we found that the percentage of black residents, percentage of Hispanic residents, and percentage of non-Hispanic white residents had GVIF values larger than 10.

## 2.4 Pre-processing

The original data set contained daily information from January 20, 2020 to December 4, 2020. We aggregated the data into months grouped by counties. There were no reported changes in the demographic information for each county because the CDC only delivered yearly data. Thus, these values stayed the same when aggregating. Next, the maximum of cases and deaths for each month was used to represent the county's cumulative count of cases and deaths. For the mean temperature, we took the average of the county's mean temperature for that month. Finally, note that the original data set reported these variables in percentages. So, we divided the values by a hundred to convert them to decimals.

After aggregation, we discovered that there were 877 missing values for mean temperature and these were eventually dropped. In addition to that, the date column was removed because it was not necessary for our research purpose. Lastly, the county column was dropped before modeling and our models would be state-level. This gave us a total of 28,955 observations with 16 continuous and 2 categorical variables.

The continuous features $x$'s, i.e., $x_1$ to $x_{14}$ in S1 Table were scaled using min-max scaling:

$$x' = \frac{x - \min(x)}{\max(x) - \min(x)}. \tag{1}$$

The min-max scaling is a commonly used scaling method. It dampened the influence of outliers and brought the range of each predictor to be between 0 and 1. Meanwhile, it would not change the shape of the distribution of each predictor, which made it easier to interpret the relationship between the $x$'s and the outcome of interest than some methods like Box-Cox transformation. For the categorical variables: "state" and "stay at home effective", they were one-hot encoded. In addition to that, we subtracted 1 from the cases column in order to apply the two-part modeling framework. A 0–1 coding strategy was used to add an indicator column showing that whether the number of cases or deaths is 0 or larger than 0: 0 represented 0 cases or deaths; 1 represented the number of cases or deaths that was larger than 0. Such a binary variable was used for the first step in the two-part modeling framework, which was explained in detail in Section 2.5.1.

Before modeling, the data set was randomly shuffled first and then split into a training set 80% vs a test set 20%.

The flowchart in Fig 3 shows how we form the final data set used in this study.

## 2.5 Models

For the purpose of comparison, we fitted an artificial neural network (ANN) model [12] and Tweedie generalized additive model (GAM) [13] without separating zeros from non-zeros. In the two-part modeling framework, which was introduced in Section 2.5.1, we first fitted a

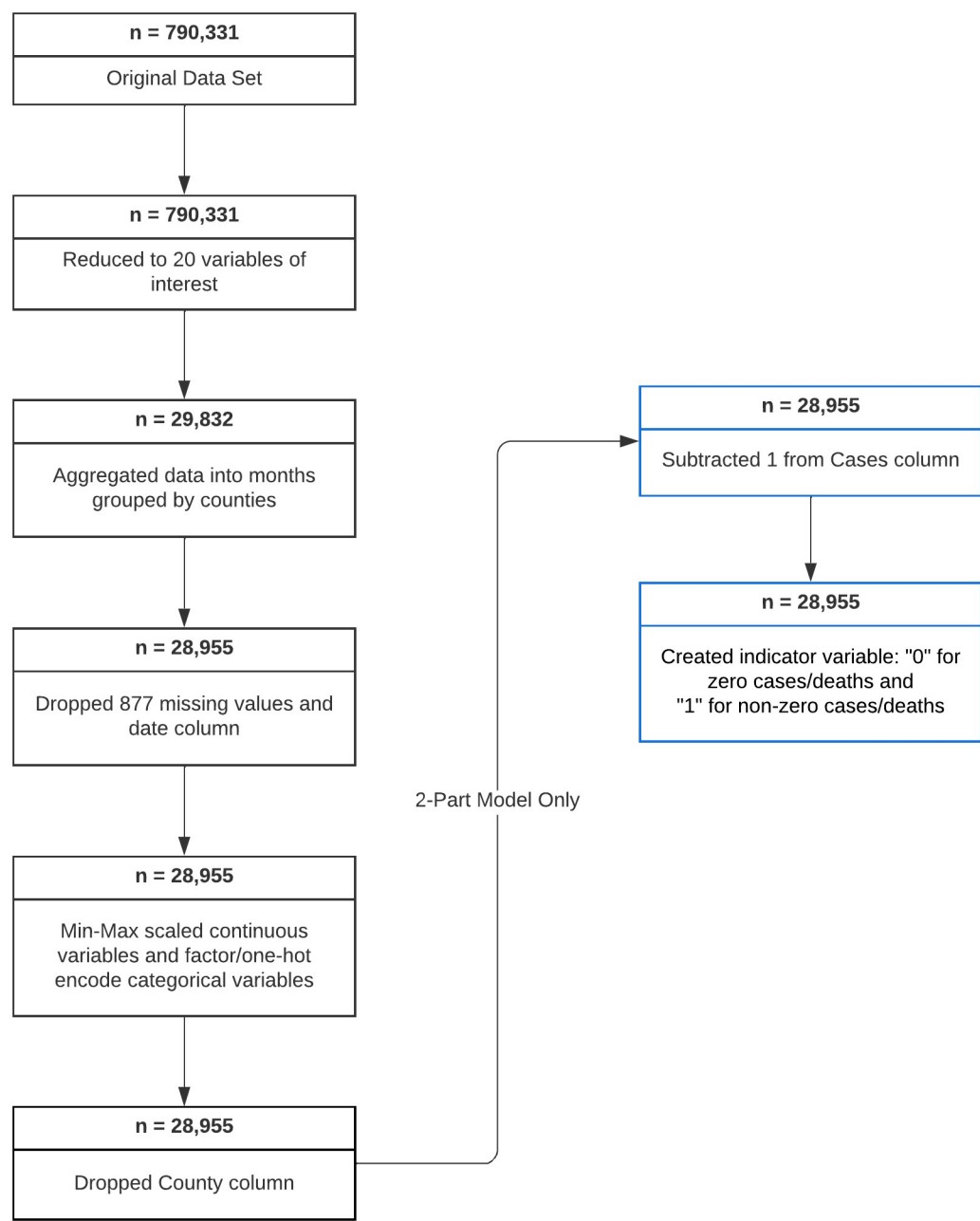

**Fig 3. Flowchart of data processing.**

logistic regression model to separate the data points into 2 groups: a group with no COVID-19 case or death and a group with at least one COVID-19 case or death. For the second group, we fitted 4 models. Background of each model was included in Section 2.5.2 to Section 2.5.5.

**2.5.1 Two-part modeling framework.** In the case of excess zeros in an outcome of interest, a two-part modeling framework could be used. The idea was to first build a binary classifier to separate data points of the outcome of interest into two parts: a part in which there was no occurrence of the outcome of interest (zero component), and a part in which at least one occurrence of the outcome of interest (non-zero component). The next step was to fit a

regression model for the non-zero component. In this step, machine learning models or statistical smoothing methods could be used.

In the first step, a logistic regression was used:

$$\log\left[\frac{p(x)}{1-p(x)}\right] = \beta_0 + \beta_1 x_{i1} + \beta_2 x_{i2} +, \ldots, +\beta_k x_{ik}, \quad i = 1, 2, \ldots, n, \tag{2}$$

where $p(x)$ is the probability of an outcome of interest is predicted to be zero or non-zero, $k$ is the total number of features, and $n$ is the total number of observations.

The logistic regression model predicted the probability of whether our response variables, cases or deaths, were zero or non-zero. If the predicted probability was above a threshold, then we would classify the response to be non-zero. We used the ROC curve [14] to find the optimal threshold value, which was the point where the weighted sum of true positive rate (TPR) and the true negative rate (TNR) were maximized. If the predicted $p(x)$ was larger than the threshold, then we classified an outcome of interest to be non-zero. For the first-step logistic model, the chosen threshold probabilities were 0.94 and 0.62 for cases and deaths respectively. The corresponding area under the receiver operating characteristic (ROC) curve values (AUC) were 0.93 and 0.95. More details were included in Section 3.1.

After separating the zero component from the non-zero component, we applied the following machine learning models: artificial neural network (ANN), random forest (RF), and extreme gradient boosting (XGBoost) models and the smoothing method: generalized additive model (GAM). Each model was explained in Section 2.5.2 to Section 2.5.5.

**2.5.2 Artificial neural network (ANN).** Artificial neural networks are computing systems that attempt to emulate neural networks in biological systems, specifically to the human brain [15]. ANNs are based on neurons, which are defined as atomic parts that compute the aggregation of their input to an output based on an activation function. An activation function defines the output of that node given a set of inputs. A single-layer ANN model can be defined as:

$$f(\mathbf{x}) = \beta_0 + \sum_{k=1}^{K} \beta_k A_k, \tag{3}$$

where $A_k = g\left[w_{k0} + \sum_{j=1}^{p} w_{kj} x_j\right]$. $K$ is the number of hidden units, $w_{kj}$'s are our weights that are adjusted through training, and $g(\cdot)$ is the activation function. Sigmoid function and ReLU function are commonly used activation functions. Here, we used the ReLU function, which is defined as $f(x) = \max(0, x)$, where $x$ is the input into the neuron. A two-layer ANN model is extended as:

$$f(\mathbf{x}) = \beta_0 + \sum_{l=1}^{K_2} \beta_l A_l^{(2)}, \tag{4}$$

where $A_l^{(2)} = g\left[w_{l0}^{(2)} + \sum_{k=1}^{K_1} w_{lk}^{(2)} A_k^{(1)}\right]$ and $A_k^{(1)} = g\left[w_{k0}^{(1)} + \sum_{j=1}^{p} w_{kj}^{(1)} x_j\right]$. The **R** package **nnet** [16] and **caret** [17] were used for tuning and fitting the ANN model except for the 2-layer model. The **python** packages **PyTorch** [18] and **scikit-learn** [19] were used for the full 2-layer ANN model.

**2.5.3 Generalized additive model (GAM).** Generalized additive models (GAM) [20] are an extension of the generalized linear model framework, where the systematic component $g[\mathbb{E}(Y)]$ can be expressed as a linear combination of unknown smooth functions of the predictors:

$$g[\mathbb{E}(Y_i)] = \beta_0 + s_1(x_{1i}) + s_1(x_{2i}) + \ldots + s_k(x_{ki}), \quad i = 1, 2, \ldots, n. \tag{5}$$

Here, $s_i$'s are smooth components fitted by penalized smoothing splines. The function $g$ is known as the "link" function connecting the population mean $\mathbb{E}(Y)$ with the linear combinations of $s_i$'s. $g$ is used to define the distribution of $\mathbb{E}(Y)$. For example, when $g$ is the identical function, $\mathbb{E}(Y)$ is assumed to follow a Gaussian or normal distribution. Such a smoothing model allows for flexibility when non-linear relationships are more complicated than a parametric quadratic or cubic form between the systematic component, which is the left-hand side of Eq (5), and predictors $x_{1i}, x_{2i}, \ldots, x_{ki}$. Tweedie distribution could be used for modeling an outcome of interest with a large positive mass at zero. As a comparison with the two-part modeling framework, we fitted a Tweedie GAM model without separating the zero chunk from the non-zero chunk. The **R** package **mgcv** [21] was used for fitting a GAM model. One advantage of a GAM model when compared with machine learning methods was its statistical inference and interpretation because of its additive nature. For example, the **R** package **mgcv** generated summary statistics such as $p$-value for each predictor. The penalty term used in each $s_i$ was chosen by generalized cross-validation.

**2.5.4 Random forest (RF).** Random forest is a decision-tree based machine learning method and can provide improvement by de-correlating the trees. Tree-based methods involve segmenting the predictors into several simple regions, which are also called nodes. A top-down approach, successive splitting, is used to divide the predictors' space into $J$ non-overlapping regions. The algorithm stops until the lowest cost, which is residual sum of squares (RSS) for a regression task, is reached. Decision trees often suffer from the over-fitting problem. Random forest improves decision trees by choosing a random subset of all candidate predictors when splitting a tree, which de-correlates the trees. Moreover, random forests are developed by fitting a number of decision trees and averaging the predicted values of each individual decision tree to make a prediction, which reduces variance of predictions [22]. The **R** package **randomForest** [23] was used to build this model in addition to the **caret** package [17], which were used to perform a 5-fold cross-validation to tune parameters in a RF model.

**2.5.5 Extreme gradient boosting model (XGBoost).** Boosting is another tree-based ensemble machine learning method. It improves the predictions of a decision tree by growing a tree sequentially, i.e., fitting a tree using residuals obtained from the previously trained trees. Each tree can be rather small with just a few terminal nodes [24]. XGBoost is a regularized form of gradient boosting and it reduces overfitting. The **R** packages **xgboost** was developed by Chen and Guestrin [25]. This package and the **caret** package were used for fitting and tuning the XGBoost model.

## 2.6 Cross validation and parameter tuning

For each model, we performed 5-fold cross-validation with the training set and computed the evaluation metrics outlined in Section 2.7 with the test set. For the ANN model, we tuned the rate $\lambda$ using gradient descent with momentum algorithm [26]. The grid for $\lambda$ to try was 0.5 and 0.1. The optimal $\lambda$ for both the full and 2-part ANN was 0.1 and 0.5 for cases and deaths, respectively. For the full ANN 2 layer model, the optimal $\lambda$ was 0.1 for both cases and deaths. For the RF model, we tuned the number of predictors randomly sampled as candidates at each split (mtry). The candidate values were integers 2 to 9. We found that the optimal mtry to be 9 predictors for both of our responses. For the XGboost model, we set $\lambda = 0.1$. Then, we tuned the subsample ratio of columns when constructing each tree (colsample_bytree), and interaction depth (max_depth). For colsample_bytree, the candidate values were 0.5, 0.6, 0.7, 0.8, and 0.9 with 10, 15, 20, and 25 for max_depth. For cases and deaths, the optimal max_depth were 10 and 15 with 0.9 and 0.8 being the optimal colsample_bytree, respectively.

## 2.7 Evaluation metrics

Because of the extreme skew pattern in the response variable, we favored evaluation metrics that were less sensitive to outliers or extreme values. In other words, we not only considered metrics such as root mean squared error (RMSE) or coefficient of determination ($R^2$) but also metrics such as mean absolute error (MAE) and quantile mean absolute error (qMAE) when choosing an optimal model. The evaluation metrics are defined as below.

**2.7.1 Coefficient of determination ($R^2$).**   The coefficient of determination or $R^2$, is the total variance explained by the model. It is defined as:

$$R^2 = 1 - \frac{RSS}{SSTO}, \tag{6}$$

where $RSS = \sum_{i=1}^{n} (Y_i - \hat{Y}_i)^2$ and $SSTO = \sum_{i=1}^{n} (Y_i - \overline{Y})^2$. $Y_i$'s are observed values, $\hat{Y}_i{'}s$ are predicted values, and $\overline{Y}$ is the mean. $R^2$ takes values between 0 and 1, inclusively. A value closer to 1 indicates a model explaining more variance in a data set.

**2.7.2 Root mean squared error (RMSE).**   The mean squared error (MSE) is the average of RSS. RMSE is obtained by taking the square root of MSE. It is a non-negative value and a value closer to 0 indicates that the overall prediction is close to the true response.

**2.7.3 Mean absolute error (MAE).**   Mean absolute error (MAE) is the average of the absolute value of prediction errors. It is less sensitive to outliers and extreme values compared to RMSE and is defined as:

$$MAE = \frac{1}{n} \sum_{i=1}^{n} \left| Y_i - \hat{Y}_i \right|. \tag{7}$$

qRMSE$_\alpha$ and qMAE$_\alpha$ are the quantile-truncated version of RMSE and MAE, respectively. To calculate these metrics, we first identified the residuals that were in the 95th percentile. These values were removed and thus, and not used for computing qRMSE$_\alpha$ and qMAE$_\alpha$. We followed the same calculations as in Duncan et al's [4].

# 3 Results

## 3.1 Optimal thresholds

As stated in Section 2.5.1, we made a ROC curve for cases and deaths respectively to obtain our optimal thresholds to separate the zero component from the non-zero component in the first step of the two-part modeling framework, which was included in Fig 4. The optimal threshold was where the true positive rate (TPR), i.e., recall and true negative rate (TNR), i.e., specificity were maximized on the ROC curve. Recall measures how many out of all positive cases, are predicted correctly. It is defined as

$$\text{Recall} = \frac{\text{TP}}{\text{TP} + \text{FN}}. \tag{8}$$

Specificity measures the proportion of correctly identified negative cases. It is defined as

$$\text{Specificity} = \frac{\text{TN}}{\text{TN} + \text{FN}}. \tag{9}$$

We found that the area under the curve (AUC) to be 0.93 and 0.95 for cases and deaths, respectively. This showed that the data was highly separable with a logistic regression model.

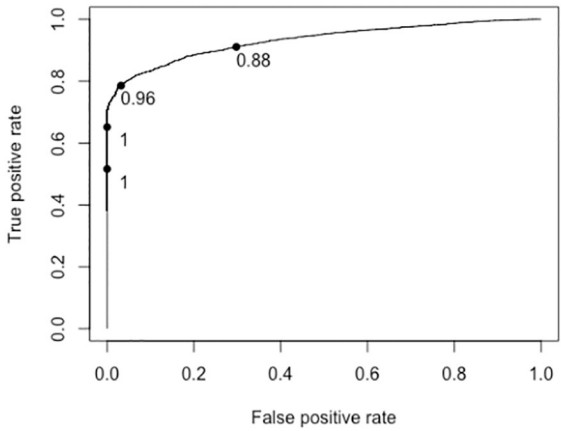 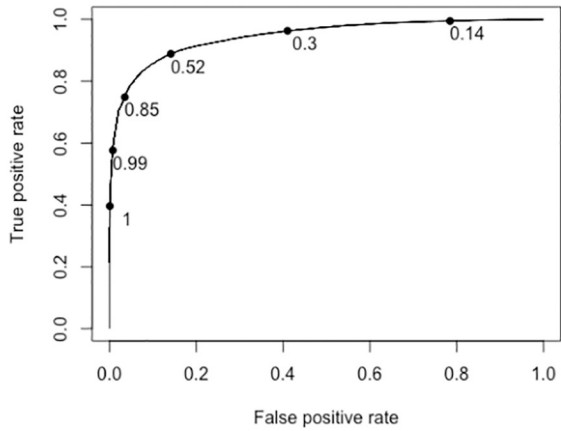

(a) ROC Curve for Cases          (b) ROC Curve for Deaths

**Fig 4. ROC curves for optimal thresholds.** Left: ROC curve for cases. Right ROC curve for deaths.

## 3.2 Model evaluation results

Tables 2 and 3 contains our model evaluation results. The two-part XGBoost model over-whelmingly outperformed the other models for both cases and deaths. The RMSE values were 584.15 and 50.57 for cases and deaths, respectively. In addition to that, 99% and 97% of the variation was explained by the two-part XGBoost model. In terms of the metrics $q\text{RMSE}_\alpha$ and $q\text{MAE}_\alpha$ which were more robust to the influence of extreme values, this model had the smallest

**Table 2. Model evaluation for positive COVID-19 cases.**

| Model | $R^2$ | RMSE | $q\text{RMSE}_\alpha$ | MAE | $q\text{MAE}_\alpha$ |
|---|---|---|---|---|---|
| Full ANN (2 layers) | 0.83 | 3712.87 | 1845.97 | 1349.57 | 59.50 |
| Full ANN (1 layer) | 0.06 | 9389.88 | 2065.60 | 2529.75 | 1507.00 |
| Tweedie GAM Model | 0.13 | 8902.26 | 1340.66 | 1270.97 | 491.71 |
| two-part Gaussian GAM | 0.86 | 3538.53 | 1457.68 | 1080.81 | 689.78 |
| two-part Random Forest | 0.84 | 3883.04 | 1564.85 | 752.99 | 398.35 |
| two-part ANN (1 layer) | 0.80 | 2179.59 | 282.17 | 456.02 | 220.12 |
| **two-part XGBoost** | **0.99** | **573.54** | **99.63** | **91.61** | **42.82** |

**Table 3. Model evaluation for COVID-19 deaths.**

| Model | $R^2$ | RMSE | $q\text{RMSE}_\alpha$ | MAE | $q\text{MAE}_\alpha$ |
|---|---|---|---|---|---|
| Full ANN (2 layers) | 0.91 | 109.48 | 48.03 | 30.65 | 4.58 |
| Full ANN (1 layer) | 0.28 | 342.65 | 72.06 | 58.00 | 29.18 |
| Tweedie GAM Model | 0.93 | 97.11 | 21.94 | 19.44 | 9.49 |
| two-part Gaussian GAM | 0.73 | 173.06 | 76.20 | 33.63 | 20.76 |
| two-part Random Forest | 0.87 | 142.40 | 66.65 | 16.57 | 8.65 |
| two-part ANN (1 layer) | 0.98 | 42.67 | 12.39 | 10.68 | 5.66 |
| **two-part XGBoost** | **0.98** | **42.65** | **3.65** | **3.21** | **1.69** |

**Table 4. Computation time comparison.**

| Model | Time (in minutes) |
|---|---|
| Full ANN (2 layers) | 0.77 |
| Full ANN (1 layer) | 16.85 |
| Tweedie GAM | 41.23 |
| two-part Gaussian GAM | 0.80 |
| two-part Random Forest | 13.01 |
| two-part ANN | 17.76 |
| two-part XGBoost | 92.07 |

metric value. Thus, we considered the two-part XGBoost model as the optimal model for both cases and deaths.

### 3.3 Computation time comparison

All models were run on a MacBook Pro with the M2 chip with 8 cores. Table 4 compares the computation time of each model when predicting deaths. Even though the two-part XGBoost model performed the best, we observed that it took the longest time to run.

### 3.4 Variable importance of the two-part XGBoost model

Fig 5 ranks features in the two-part XGBoost model according to their feature importance value, which was determined by how much the squared prediction error over all trees was reduced as a result of including the feature. For cases as our response, two-part XGBoost found deaths to be the most important feature and population to be the second most important feature; for deaths, the model found cases and the rate of primary care physicians to be the top two important features.

### 3.5 Interpretation of the Tweedie GAM model

When modeling deaths, although the Tweedie GAM model did not have the best evaluation metric values, its $R^2$ value was 93% and its truncated RMSE value was comparatively low. For the Tweedie GAM model predicting deaths, the continuous predictors were all significant with a $p$-value smaller than 0.05, and the categorical predictors were also significant except for some

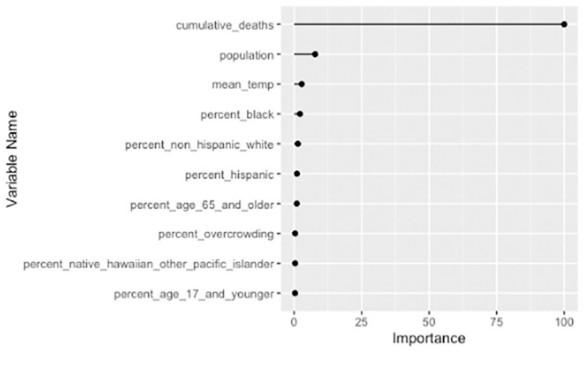

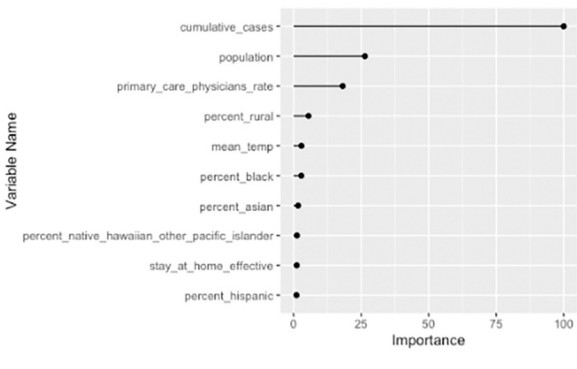

(a) Cases                                   (b) Deaths

**Fig 5. XGBoost feature importance ranking.** Left: feature importance ranking for cases. Right: feature importance ranking for deaths.

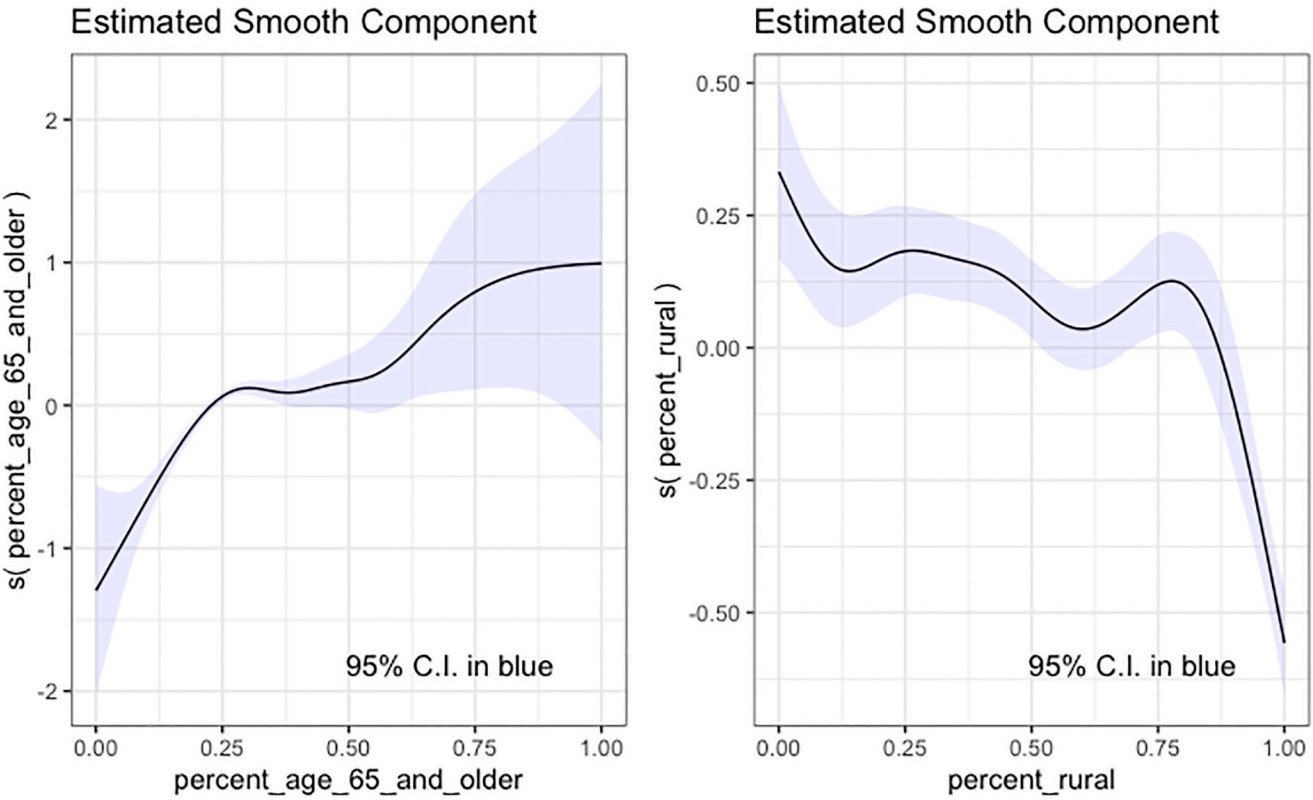

**Fig 6. Two smooth components estimated by a Tweedie GAM model predicting deaths.** Left: the estimated smooth component for the feature percentage of people older than 65. Right: the estimated smooth component for the feature the percentage of people living in rural communities.

states such as Arizona, Maine, etc. In Fig 6, we show two estimated smooth components. In the Tweedie model, a logarithm link function was used, and we could interpret the smooth components as when the percentage of people older than 65 increased, more deaths were expected, and the increasing rate slowed down when the percentage was close to 1; as the percentage of people living in rural communities increased, less deaths were expected, and the decreasing rate increased when the percentage was close to 1.

### 3.6 Sensitivity analysis with the XGBoost model

To enhance the findings of the XGBoost model, we did 2 sets of sensitivity analysis using the **sensitivity** package in Python [27]. The predictors we chose to vary were the top 3 predictors in the variable importance ranking plots shown in Fig 5. First, we considered cases as the response variable and chose deaths, population, and mean temp as the 3 predictors to vary. Specifically, we chose 0 and another 4 quantiles in the non-zero component in deaths, chose 5 quantiles in population, and 5 quantiles in mean temp. For other predictors, we fixed it at its mean value if the predictor is continuous; if the predictor was categorical, then we fixed the predictor at the category with the most observations. This gave us 125 combinations and we evaluated the XGBoost model with each of the 125 combinations. Heatmaps were created for cases vs deaths, cases vs population, and cases vs mean temp. In the second set of sensitivity analysis, we considered deaths as the response variable, and chose 0 and another 4 quantiles in

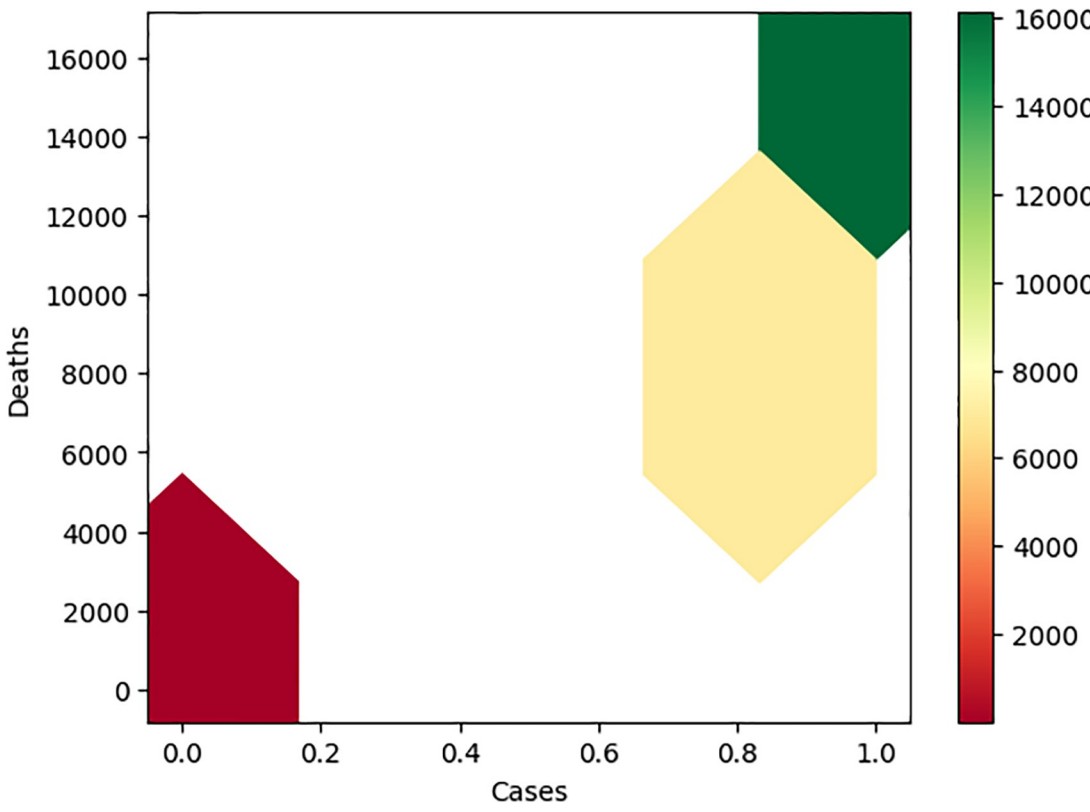

**Fig 7. Sensitivity analysis heatmap of COVID-19 deaths vs COVID-19 cases.**

the non-zero component in cases, chose 5 quantiles in primary care physicians rate, 5 quantiles in population. Other parts were the same as the first set of sensitivity analysis.

The information in the heatmap included in Fig 7 is similar to that shown in Fig 2. However, in this heatmap, we saw that as the number of COVID-19 cases was large, most numbers of deaths were predicted to be around the median value of the distribution of deaths. Another five heatmaps were included in Supporting information.

## 4 Conclusion

In this paper, we proposed a two-part modeling framework for a large integrated COVID-19 data set with 28,955 observations and 18 variables from Kaggle at https://www.kaggle.com/datasets/johnjdavisiv/us-counties-covid19-weather-sociohealth-data. In the first part, the framework used a logistic regression model to separate the zero component from the non-zero component. For both cases and deaths, the AUC value was high, i.e., 0.93 and 0.95, which meant that the model distinguished the two components well. In the second part, generalized additive model and machine learning methods: random forest, gradient boosting, and artificial neural network were used to fit the distribution of the non-zero observations. Through a 5-fold cross-validation, each method's performance was evaluated. We presented evidence that a two-part XGBoost model had the best evaluation metric values, across the board when cases and deaths were the response variables. We also found that the one-layer ANN model performed much better when used in a two-part framework. Such kind of a two-part modeling

framework worked well for the data set. Data sets, not only COVID-19 data, which have excess zeros could potentially be well fitted with this framework.

One limitation we encountered with our data set was the complexity of the county variable. Since we randomly sampled our training set and test set, the test set did not include the same number of factors found in the training set. Thus, we had to drop the county column when building models. Note that stratified $k$-fold cross-validation was not our preferred sampling method [28]. It is designed to deal with an imbalanced target outcome of interest for a classification task. This type of sampling is costly, especially when applied to large data sets. It would be too computation-wise costly to use this method to address the issue with our imbalanced county variable. Another limitation was that it took a long time to train the two-part XGBoost model. The XGBoost open-source software by Chen and Guestrin [25] provides parallel computing. In the future, this model could be improved with parallel computing such that it will take less time to run.

## Supporting information

**S1 Table. Summary of continuous variables.**
(DOCX)

**S2 Table. Summary of categorical variables.**
(DOCX)

**S1 Fig. Sensitivity analysis heatmap of COVID-19 cases vs COVID-19 deaths.**
(TIFF)

**S2 Fig. Sensitivity analysis heatmap of COVID-19 cases vs population.**
(TIFF)

**S3 Fig. Sensitivity analysis heatmap of COVID-19 deaths vs population.**
(TIFF)

**S4 Fig. Sensitivity analysis heatmap of COVID-19 cases vs mean temperature.**
(TIFF)

**S5 Fig. Sensitivity analysis heatmap of COVID-19 deaths vs primary care physicians rate.**
(TIFF)

## Author Contributions

**Data curation:** Teresa-Thuong Le.

**Formal analysis:** Teresa-Thuong Le.

**Investigation:** Teresa-Thuong Le.

**Methodology:** Xiyue Liao.

**Software:** Teresa-Thuong Le.

**Supervision:** Xiyue Liao.

**Validation:** Teresa-Thuong Le.

**Visualization:** Teresa-Thuong Le.

**Writing – original draft:** Teresa-Thuong Le.

**Writing – review & editing:** Xiyue Liao.

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
