## [Decision Letter · Decision Letter 0]

24 Jan 2024

PONE-D-23-42948Two-part predictive modeling for COVID-19 cases and deaths in the U.S.PLOS ONE

Dear Dr. Liao,

Thank you for submitting your manuscript to PLOS ONE. After careful consideration, we feel that it has merit but does not fully meet PLOS ONE’s publication criteria as it currently stands. Therefore, we invite you to submit a revised version of the manuscript that addresses the points raised during the review process.

We look forward to receiving your revised manuscript.

Kind regards,

Jyotir Moy Chatterjee, M. Tech (CSE)

Academic Editor

PLOS ONE

Journal Requirements:

3. Please note that your Data Availability Statement is currently missing the direct link to access each database. If your manuscript is accepted for publication, you will be asked to provide these details on a very short timeline. We therefore suggest that you provide this information now, though we will not hold up the peer review process if you are unable.

**Additional Editor Comments:**

Major Revision

Reviewers' comments:

Reviewer's Responses to Questions

**Comments to the Author**

1. Is the manuscript technically sound, and do the data support the conclusions?

Reviewer #1: Yes

Reviewer #2: Yes

2. Has the statistical analysis been performed appropriately and rigorously? 

Reviewer #1: N/A

Reviewer #2: Yes

3. Have the authors made all data underlying the findings in their manuscript fully available?

Reviewer #1: No

Reviewer #2: Yes

4. Is the manuscript presented in an intelligible fashion and written in standard English?

Reviewer #1: Yes

Reviewer #2: Yes

5. Review Comments to the Author

Reviewer #1: The paper illustrates a two-step approach to provide predictions of cases and deaths concerning COVID-19 data referred to US. The authors propose using logistic regression and standard machine learning models. Comparisons among predictive models are based on standard metrics.

among predictive models are based on standard metrics.

The authors have to account for the minor and major concerns raised in the attached pdf file.

Reviewer #2: The article provides a compelling presentation of a prediction model, showcasing clear methodology. The objective of this paper is to present a predictive model, consisting of two parts, for anticipating the occurrences of COVID-19 cases and deaths. Different machine learning and statistical methods are examined and their performances are compared with each other in this paper.

The study fits into the journal’s scope. The claims properly placed in the context of the previous literature and the literature review is well-organized. The English writing requires editing and proofreading. The following are my major and minor comments.

Major Comments

1. It appears that there might not be explicit mention of checking for collinearity, for example, through methods like the Variance Inflation Factor (VIF). Assessing collinearity can be crucial in ensuring the robustness of regression models.

2. Given that your model incorporates population, total infected cases, and total deaths, it appears these variables may have interdependencies. To enhance the robustness of your findings, it is valuable to conduct a sensitivity analysis. This would allow you to explore how changes in variables may impact the model results and provide insights into the stability of your conclusions.

3. The manuscript must be checked for punctuation, and errors in grammar .

Minor Comments

• There should be no citation in the abstract section according to journal’s guidelines. It seems you cited your data source in the abstract.

• The authors have used the words ”COVID-19” and ”COVID” inconsistently. I think it is better to use ”COVID-19” throughout the papaer.

• Line 142: there is a small typo. ”as least” should be corrected to ”at least”.

• Line 192: there is a small punctuation error. The punctuation after ”function” should be corrected to ”,”.

• Line 267: there should be ”.” after ”Fig 4”.

• I noticed that present tense was consistently used in the Results and Materials and Methods sections. For example in line 231: ”We find that the optimal mtry to be 9 predictors for both of our responses” or in line 271: ”We find that the area under the curve (AUC) to be 0.93 and 0.95 for cases and deaths, respectively.”. It’s generally recommended to use the past tense when describing findings.

• I noticed that while some equations in the manuscript are numbered, there are instances where equations are not assigned numbers. I recommend ensuring consistent equation numbering throughout the article.

• Line 142: there is a small typo in ”Fig 5 rank features in the two-part XGBoost model according to their feature importance value”. ”rank” should be corrected to ”ranks”.

• line 296: ”One advantage of a GAM model when compared with machine learning methods is its statistical inference and interpretation. For example, the R package mgcv will generate summary statistics such as p-value for each predictor.” should be moved to Materials and Methods Section.

6. PLOS authors have the option to publish the peer review history of their article (what does this mean?). If published, this will include your full peer review and any attached files.

Reviewer #1: No

Reviewer #2: No

---

## [Author Response · Author response to Decision Letter 0]

6 Mar 2024

The responses to reviewers' comments are submitted in the "Attach Files" section as a .pdf file

---

## [Decision Letter · Decision Letter 1]

2 Apr 2024

Two-part predictive modeling for COVID-19 cases and deaths in the U.S.

PONE-D-23-42948R1

Dear Dr. Liao,

We’re pleased to inform you that your manuscript has been judged scientifically suitable for publication and will be formally accepted for publication once it meets all outstanding technical requirements.

Kind regards,

Jyotir Moy Chatterjee

Academic Editor

PLOS ONE

Additional Editor Comments (optional):

Reviewers' comments:

Reviewer's Responses to Questions

**Comments to the Author**

1. If the authors have adequately addressed your comments raised in a previous round of review and you feel that this manuscript is now acceptable for publication, you may indicate that here to bypass the “Comments to the Author” section, enter your conflict of interest statement in the “Confidential to Editor” section, and submit your "Accept" recommendation.

Reviewer #1: All comments have been addressed

Reviewer #3: All comments have been addressed

2. Is the manuscript technically sound, and do the data support the conclusions?

Reviewer #1: Yes

Reviewer #3: Yes

3. Has the statistical analysis been performed appropriately and rigorously? 

Reviewer #1: Yes

Reviewer #3: Yes

4. Have the authors made all data underlying the findings in their manuscript fully available?

Reviewer #1: Yes

Reviewer #3: Yes

5. Is the manuscript presented in an intelligible fashion and written in standard English?

Reviewer #1: (No Response)

Reviewer #3: Yes

6. Review Comments to the Author

Reviewer #1: The authors addressed all my concerns. They added some relevant features and improved the paper's quality.

It is suitable for publication.

Reviewer #3: The author responded and made revisions to the review comments one by one. The revised article meets the publication standards and I believe it is acceptable.

7. PLOS authors have the option to publish the peer review history of their article (what does this mean?). If published, this will include your full peer review and any attached files.

Reviewer #1: No

Reviewer #3: No
